# Lipid-Based Inhalable Micro- and Nanocarriers of Active Agents for Treating Non-Small-Cell Lung Cancer

**DOI:** 10.3390/pharmaceutics15051457

**Published:** 2023-05-10

**Authors:** Sona Gandhi, Indrajit Roy

**Affiliations:** 1Department of Chemistry, School of Basic & Applied Sciences, Galgotias University, Greater Noida 203201, India; gandhi7hd@gmail.com; 2Department of Chemistry, University of Delhi, Delhi 110007, India

**Keywords:** non-small-cell lung cancer, liposomes, solid lipid nanoparticles, lipid micelles, inhalable drug delivery, chemotherapy, gene therapy

## Abstract

Non-small-cell lung cancer (NSCLC) afflicts about 2 million people worldwide, with both genetic (familial) and environmental factors contributing to its development and spread. The inadequacy of currently available therapeutic techniques, such as surgery, chemotherapy, and radiation therapy, in addressing NSCLC is reflected in the very low survival rate of this disease. Therefore, newer approaches and combination therapy regimens are required to reverse this dismal scenario. Direct administration of inhalable nanotherapeutic agents to the cancer sites can potentially lead to optimal drug use, negligible side effects, and high therapeutic gain. Lipid-based nanoparticles are ideal agents for inhalable delivery owing to their high drug loading, ideal physical traits, sustained drug release, and biocompatibility. Drugs loaded within several lipid-based nanoformulations, such as liposomes, solid-lipid nanoparticles, lipid-based micelles, etc., have been developed as both aqueous dispersed formulations as well as dry-powder formulations for inhalable delivery in NSCLC models in vitro and in vivo. This review chronicles such developments and charts the future prospects of such nanoformulations in the treatment of NSCLC.

## 1. Introduction

Lung cancer is the leading cause of cancer deaths for all sexes worldwide owing to its high incidence and low survival rates [1]. This disease is both inherent (in people with a family history of lung cancer), as well as acquired through various lifestyle, occupational and environmental factors such as smoking (active and passive), prolonged inhalation of toxic gases, and occupational exposure to hazardous chemicals/carcinogens such as heavy metals, asbestos, etc. [2,3]. According to the latest statistics, about 85% percent of all lung cancers are non-small-cell lung cancers (NSCLCs), which are further categorized into adenocarcinoma, squamous cell carcinoma, and large cell carcinoma. Despite the various etiologies, NSCLC is largely untreatable with existing surgical and chemo-radiation therapeutic interventions and remains a leading cause of death worldwide [4]. This could be attributed to the fact that NSCLCs tend to metastasize early and develop resistance to a large variety of anticancer drugs [5].

Only a small proportion of NSCLC patients, with only localized solid tumors at the time of diagnosis, are amenable to surgery and/or radiation therapy. However, most patients are diagnosed at a late stage, when the cancer has extensively metastasized to other parts of the lungs. For such patients, systemically administered chemotherapy remains the only treatment avenue, although several new experimental therapies, such as immunotherapy, are being explored as the standard of care against lung cancer. In general, several drugs, such as cis-platin, paclitaxel, 5-fluorouracil, etc., have been delivered intravenously in NSCLC patients [6,7]. However, as most of these drugs are poorly water soluble, they have been delivered using non-biocompatible excipients, such as castor oil [8]. Intravenously administered drug formulations face several hurdles before they reach their intended target (NSCLC sites in this case). First, these formulations have a low circulation half-life and poor pharmacokinetics/pharmacodynamics (PK/PD). Even for the proportion of drugs that attain stable systemic circulation, for effective permeation to NSCLC sites, they have to cross the blood–air barrier, which prevents the pulmonary entry of most of the circulating drugs. There are several other challenges to overcome as well, which include the non-targeted nature of these therapies, multi-drug resistance shown by these cancers, poor uptake in cancer cells, immunogenicity, toxicity to normal cells/tissues, etc. [9]. These challenges necessitate the administration of very high as well as multiple dosages of toxic drugs [10]. The toxicities of chemotherapeutic drugs, and/or the non-biocompatibility of the excipients, lead to several adverse effects, such as hair loss, nausea, vomiting, trauma, breathing difficulties, and immunogenic reactions [11]. As a result, the clinical outcome of these therapies is largely dismal.

Over the past few decades, the use of nano- and microparticulate carriers for the delivery of drugs to target sites, especially in cancer, has witnessed unprecedented benefits [8,12,13]. These carriers render stable aqueous dispersion of poorly water-soluble drugs, protect them from physiological degradation, enhance their bioavailability, help them traverse safely through biological barriers, allow efficient entry inside target cells and intracellular organelles, and release drugs in a sustained manner. In addition to chemotherapeutic agents, other active components, such as proteins, peptides, DNA, RNA, etc., can be carried and delivered using these carriers. Finally, one or more imaging probes can be integrated along with therapeutic components with these carriers, which allows image-guided therapeutics and/or real-time monitoring of therapy. Target specificity can be achieved either passively, via the enhanced permeation and retentivity (EPR) effect or actively, via surface grafting with ligands that recognize specific receptors overexpressed on tumor tissues/cells/sub-cellular organelles [14]. However, the EPR effect, which is based on the higher extravasation of circulating macromolecules in tumor tissues owing to their leaky vasculature, as well as their poor lymphatic drainage, has been found to be more pronounced in small animals than in humans [15]. This is a key reason behind the poor clinical translation of several antitumor drug delivery strategies found to be successful in small animal studies. One or more active therapeutics stably incorporated within such micro/nanocarriers can be administered in the body via various routes, such as intravenous, intraperitoneal, transdermal, oral, ocular, intranasal, intratumoral, etc. The physical characteristics of such drug carriers can be tuned to suit a particular route of delivery. For example, for intravenous administration, the particles should be ultrasmall (a diameter below 100 nm), with a hydrophilic and neutral surface, and preferably degradable in an acidic environment. On the other hand, for oral delivery, the carrier size can be larger (e.g., microparticles) but should withstand the high acidic microenvironment of the stomach cavity.

In the context of drug delivery to the lungs, the direct, inhalable route of administration has several perceived benefits over systemic, intravenous administration [16]. Direct delivery is less invasive and more patient-friendly, requires a low drug dosage, and minimally affects non-target sites [17]. This mode of delivery has primarily evolved for the treatment of two major pulmonary disorders, namely asthma and chronic obstructive pulmonary disorder (COPD), where anti-inflammatory corticosteroids and bronchodilators are routinely delivered in simple, non-clinical settings. Nevertheless, this delivery method is equally effective for treating various lung cancers, including NSCLC.

Various micro/nanocarriers incorporating chemotherapeutic or other anticancer agents can be stably formulated for inhalable delivery. There are certain parameters that should be taken into consideration before designing carriers for inhalable drug delivery: size, shape, and surface functionality. These factors greatly influence the deposition of the drug in the lungs and their bioavailability. The aerosol particles can be categorized into three groups based on their size: coarse (>2 μm), fine (0.1–2 μm), and ultrafine (<0.1 μm) [18]. The site and mechanism of deposition can be altered by changing the particle size. Particles smaller than 0.5 μm are deposited via diffusion in the alveoli [19]. The particles in the size range of 1–5 μm are deposited via sedimentation in the small airways of the bronchioles and alveoli. Hygroscopic particles are known to grow while passing through moist and warm air passages and are mainly deposited via sedimentation [20]. The bigger particles (size greater than 5 μm) are deposited via impaction, which depends a lot on the aerodynamic diameter and mass. This is the common mechanism for the deposition of drugs using dry-powdered formulations [21]. The particles in the size range 1–5 μm are deposited in the small airways and alveoli. The size of the particles can be fine-tuned as per the requirement to deposit the drug at the targeted site in the respiratory pathway.

The shape of the particles also plays a key role in influencing the clearance by alveolar macrophages. Orientation is also important; phagocytosis occurs very slowly when macrophages are attached to the minor axis of particles with shapes such as elliptical disks and rectangular disks. In the case of the spheres, the shape is symmetrical, and particles are engulfed by the macrophages irrespective of the point of attachment [22]. Since the commonly used lipid micro/nanoparticles are spherical, the shape is not a controlling parameter. The surface functionality of the particles can control drug absorption and degradation, along with pulmonary clearance. This, in turn, can control the drug’s half-life in the lungs. It is preferred that the surface should contain moieties that can provide the particles with stealth properties, which can prevent biofouling and the immune response [23]. In a study, it was observed that when drug-loaded microspheres were coated with dipalmitoylphosphatidylcholine (DPPC), the macrophage uptake of the active ingredient was reduced [24]. Even though the inhalable route directly accesses the lungs, it does encounter the mucosal barrier, which is a dense, hydroscopic layer of proteins (mucins) that prevents the entry of undesirable particles into the lungs [17]. The surface properties of micro/nanocarriers play a critical role in passing through this barrier. Mucins are anionic and also contain hydrophobic components. Therefore, carriers with cationic surface charge or lipophilic surfaces tend to adhere to the mucosal barrier via electrostatic or hydrophobic interactions, respectively, and are not able to effectively traverse through this barrier and reach the lungs. Therefore, carriers with hydrophilic surface and anionic/neutral surface charge are desirable for efficient delivery to the lungs. Among the various types of carriers, lipid-based micro/nanocarriers are ideally suited for inhalable delivery owing to their superior aerodynamics, which allows higher retention and low drug loss following delivery. In this review, we shall focus on the use of lipid-based micro/nanoparticulate carriers for the inhalable delivery of active agents for the treatment of NSCLC. First, we shall highlight the principal strategies and equipment employed for inhalable drug delivery to the lungs. Next, we shall discuss the various lipid-based formulations used for the incorporation and delivery of active therapeutic agents. Following that, we shall present selected examples of lipid-based micro/nanoparticles used for the inhalation delivery of not only chemotherapeutic drugs but also other active agents, such as genetic molecules, in NSCLC models in vitro and in vivo. Finally, we shall provide conclusions and future directions for this growing biomedical discipline.

## 2. Various Types for the Delivery of Inhalable Therapeutics

Therapeutic formulations can be delivered to the lungs via the inhalation route using different devices/equipment, as depicted in Figure 1. The common ones that can be used for pulmonary delivery to NSCLC are (i) nebulizers; (ii) dry powder inhalers (DPI), and (iii) pressurized meter dose inhalers (pMDI). Nebulizers convert liquid-dispersed formulations into tiny droplets, which can be inhaled. However, nebulizers require electricity/a battery for operation. Furthermore, the shear force associated with nebulizers can lead to damage to sensitive therapeutic agents, such as RNA molecules. On the other hand, dry nanotherapeutics are delivered either with the use of mechanical propellers (dose inhalers), or simply as dry-powder inhalers (DPI), both of which do not require any external power supply (e.g., electricity) [25]. Here, the aerosolized drug is given through a mouthpiece into which the patient can inhale, and the drug reaches the lungs. DPIs work with solid formulations and no propellants are required. The drug is inhaled in proportion to the inspiratory pull of the patient [26]. In contrast, pMDI uses a canister of the drug at high pressure contained inside a plastic tube along with a mouthpiece. Upon spraying, the equipment gives a precise and consistent dosage of the drug [27].

Nebulizers are the most widely used devices for pulmonary delivery during clinical studies. They can be used to deliver liquid aerosol in the form of very fine droplets and are hence used to work with simple formulations such as solutions and suspensions. There are several types of nebulizers available: ultrasonic, vibrating mesh, and jet nebulizers [28]. These are especially beneficial in cases where active inhalation or mechanical ventilation is not feasible [29]. Even though nebulizers are the most widely used equipment they are not the most efficient device for pulmonary delivery. The administration of drugs via nebulization needs multiple cycles and takes time. The lung deposition achieved is also unsatisfactory; this is due to the fact that a major amount of the formulation is not inhaled and is lost in the device and surroundings; only about 10–15% of the total drug is finally deposited in the lungs [30]. Another challenge is that for effective nebulization the drug must be water soluble, which is not the case with most lipid-based formulations. Moreover, the process of nebulization can impact the size, the drug loading capacity of the carrier, and the release profile of the drug. The long-term stability of drugs is poor in the case of liquid formulations as compared to dry ones, so the long-term stability of drugs is somewhat compromised in the case of nebulization [31]. The dry powder formulation is better than nebulization in the above-mentioned concerns. DPIs are quite easy to use and can even be self-administered at home. They are also economical to use and easily transportable and can deliver high doses of anticancer drugs to the lungs. The long-term stability is better as DPI works with solid formulations; in addition, it does not have the pre-requisite of water solubility for the formulations [32]. However, there is a shortcoming with this approach as well. If the adhesive and cohesive interactions are high, then the drug sticks to the carrier and to the device which reduces the overall efficiency [33].

The dosage is different in different cases, so the device is chosen depending on this. When the required dosage is small (about a few micrograms), then pMDIs are used. On the other hand, nebulizers are used to deliver larger doses [34]. They can also be chosen based on their aerodynamic sizes; this is because the dynamic properties of any formulation inside the airway depend a great deal on its size and shape. Thus, formulations within the size range 5–1000 nm, such as micelles, liposomes, and SLNs, are administered with the help of a nebulizer. Formulations with sizes greater than 5 m, such as microparticles and nanocomposites, are administered using DPIs.

**Figure 1 pharmaceutics-15-01457-f001:**
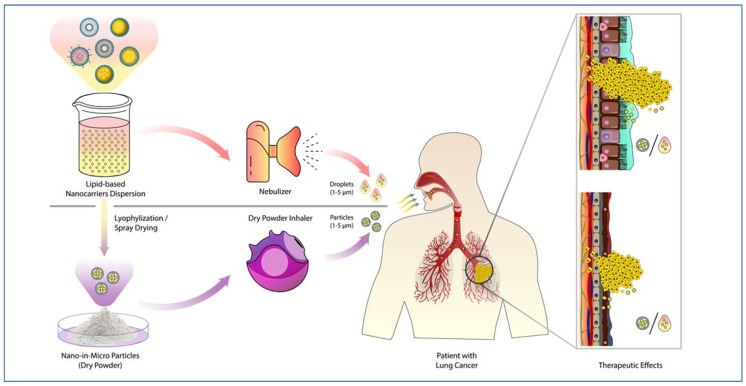
Main types of inhalable drug delivery [35], copyright (2021) @ MDPI.

## 3. Lipid-Based Micro/Nanomaterials as Drug Carriers: Various Types

The concept and design of synthetic lipid-based drug delivery vehicles is in-spired by the various naturally occurring lipid-containing micro/nanocarriers or their constituents in the body, such as whole cells, intracellular or extracellular vesicles, cellular membranes, lipid rafts, etc. Lipid-based drug delivery vehicles have gained much attention lately due to their high biocompatibility, site-specificity, and controlled drug release. Lipid-based microparticles and nanoparticles are particles that are usually spherical and have at least one lipid bilayer enclosing at least one aqueous compartment. When it comes to delivery systems, lipids offer several advantages, namely self-assembly, ease of formulation, large payload capacity, and easily modifiable physicochemical properties [36,37]. Owing to these properties, lipid-based formulations happen to be amongst the most common FDA-approved nanomedicines [38], with the names of such drugs being mentioned in Table 1 [39,40,41,42].

### 3.1. Liposomes

These nanoparticles are usually made up of phospholipids that form unilamellar and multilamellar vesicular structures. This enables liposomes to carry hydrophobic as well as hydrophilic drugs. In fact, both can be entrapped within the same nano-formulation which enhances its usability [43]. The hydrophilic drugs are loaded in the cavity and the hydrophobic ones are embedded within the phospholipid bilayer. Their stability during in vitro and in vivo experiments can be fine-tuned by controlling their synthetic procedure, which can alter their size, composition, number of layers, surface charge, and functionality [44]. Phosphatidylcholines (PC), phosphatidylethanolamine (PE), phosphatidylserines (PS), phosphatidylglycerol (PG), and sphingomyelin are the most widely used phospholipids for liposome synthesis [45]. The net charge carried by liposomes depends on the type of phospholipid, which can be neutral, cationic, or anionic. The neutral ones are more commonly used, as the charged ones are cleared faster from the system due to their interactions with opsonizing proteins [46]. When unsaturated egg or soybean PCs are used as components, the liposomes are less stable and extremely permeable, whereas saturated phospholipids containing long acyl chains promote rigidity and stability with an impermeable layer [47].

Liposomes are classified on the basis of their size and the number of lipid bilayers. Unilamellar vesicles contain only one bilayer and multilamellar vesicles consist of more than one bilayer arranged in a layered structure, like onion separated by water, with a size less than 500 nm. Unilamellar vesicles can be further divided into small unilamellar vesicles (SUVs, 20–100 nm) and large unilamellar vesicles (LUVs, >100 nm). The drug loading capacity is controlled by the size and number of bilayers. The circulation time also varies with the particle size; liposomes with a size greater than 100 nm are more susceptible to opsonization and removal by the reticuloendothelial system (RES) [48]. The problem of clearance by the RES can be resolved by modifying the liposome’s surface, which can help by extending the circulation time and enhancing delivery, which is essential for clinical usage [49].

### 3.2. Lipid Nanoparticles (LNPs)

These are similar to liposomes and have been used for delivering nucleic acids. They are slightly different from liposomes as they can form micelles or micellar structures inside the particle core. This morphology is fairly easy to alter by changing the formulation and synthesis parameters [50]. LNPs are made up of four major components: (i) ionizable or cationic lipids (which can conjugate with negatively charged nucleic acids), (ii) phospholipids, for the structural integrity of the particle, (iii) cholesterol to impart stability and fusion with target cell membranes; and (iv) PEGylated lipids to enhance stability and circulation [51]. Other functionalities can also be attached to LNPs to promote their targeting ability to specific organs, such as the brain [52]. LNPs have been well explored for the purpose of oral, parenteral, ocular, and topical drug delivery [53]. These are economically viable to fabricate and promising alternatives to most colloidal and micron-sized delivery systems, as they do not require any organic solvent during synthesis and the component lipids are GRAS (generally regarded as safe). Both small molecular and macromolecular drugs can be attached to LNPs. Two types of LNPs with solid matrix have been reported. The first generation of lipid nanoparticles, or solid lipid nanoparticles (SLNs), are made up of triglycerides, which have a high melting point and are solid at room and physiological temperatures. Hydrophobic and hydrophilic drugs can be incorporated into the lipid melt by using techniques such as microemulsion, membrane contactor, coacervation, phase inversion temperature, double emulsion, solvent diffusion, emulsification solvent evaporation, ultrasonication, and several types of homogenization [54]. The second generation of lipid nanoparticles, or nanostructured lipid carriers (NLCs), are composed of a less-ordered crystalline structure of the lipid matrix, which promotes higher drug loading capacity [55].

Lipids with long-chain fatty acids are used for the fabrication of LNPs, whereas lipids containing medium-chain fatty acids or unsaturated fatty acids are used as liquid lipids for the fabrication of NLCs [56]. SLNs have a somewhat low loading capacity for hydrophilic drugs; to improve upon this, hydrophobic ion pairing can be carried out [57]. The added ion can interact and form ion pairs with the charged hydrophilic drug molecule. One more approach is to fabricate polymer–lipid conjugates; the hydrophilic drugs can easily interact with and attach to the polymer part [58]. SLNs also have a lower drug-carrying capacity as compared to NLCs due to the tightly packed lipid matrix. On the other hand, NLCs have a less densely packed lipid matrix which ensures that there is more empty space available for drug loading. Several studies have been carried out to explore the usage of lipid nanoparticles as drug delivery vehicles [59,60].

A special category of lipid nanoparticles has emerged in recent times; these are naturally derived lipid nanoparticles from fresh fruits and vegetables and are commonly termed plant-derived lipid nanoparticles (PDLNPs). Different types of PDLNPs have been explored and they have shown targeting ability upon oral administration [61]. They have a structure similar to liposomes comprising an aqueous core and a lipid bilayer. The difference is that the bilayer in the case of PDLNPs is loaded with glycolipids but lacks cholesterol. Since cholesterol is known to impart stability, PDLNPs should have greatly compromised stability. However, it has been studied in several cases that PDLNPs were more stable under physiological conditions as compared to the corresponding liposomes [62]. Several PDLNPs have shown anticancer activity, which can be attributed to the presence of natural active ingredients of plant origin, such as the fact that there are active components present in ginger-derived nanoparticles [63]. These can be further modified to enhance the targeting efficacy, by allowing the lipids extracted from PDLNPs to self-assemble in a buffer solution and then attaching drug cargo and targeting moieties [64].

### 3.3. Lipid Micelles

These are amphiphilic lipids that can self-assemble into micellar structures in aqueous medium. Micelles have a hydrophobic core and have been studied extensively for drug delivery applications for hydrophobic drugs. Lately, efforts have been made to improve drug loading capacity and stability and reduce micelle–cell interaction during circulation. PEGylation is one of the promising options to overcome such shortcomings [65]. These micelles can easily be grafted with targeting moieties and efficiently entrap therapeutic agents and are hence usable in the treatment of several diseases, particularly cancer. Several studies have explored the efficacy of lipid micelles for drug delivery [66].

### 3.4. Lipid Nanodiscs and Nanocubosomes

Nanodiscs are disc-shaped particles within the size of 50 nm and contain a membrane made up of lipids and a belt made of peptides or other polymers, which holds the disc together. This structure is conducive to delivering membrane proteins, hydrophobic drugs, and protein–drug conjugates that help with combination therapy. The polymer belt can be modified and functionalized to promote its targeting ability [67]. Cubosomes are composed of liquid crystalline particles that are composed of a lipid cubic phase, which are then stabilized by an outer covering of a polymer [68]. The lipid cubic phase is made by the self-assembly of amphiphilic lipids in the presence of a fixed amount of water. These structures have a peculiar feature wherein a continuous lipid bilayer is present with curved water channels which facilitates the entrapment of a variety of drugs, hydrophilic as well as hydrophobic. The lipid composition controls the pore size, and the outer coating of the polymer controls the targeting ability [69]. These nanoparticles fabricated by breaking down large-sized cubic-phase particles have a large surface area-to-volume ratio, which enhances their protein binding ability as compared to liposomes of comparable size [70]. These nanocubosomes have a lower viscosity than the regular ones and are more stable in physiological conditions [71]. Hence, they hold a lot of potential for their usage in drug delivery. Both nanodiscs and nanocubosomes have been explored for their drug delivery potential via various studies [72,73].

### 3.5. Naturally Occurring Lipid Micro/Nanocarriers

In addition to the synthetic lipid-based drug carriers mentioned in the above sections, several naturally occurring lipid-containing vesicles can be isolated from body fluids and ex vivo engineered/reprogrammed for a particular diagnostic and/or therapeutic intent. The bio-compatibility and naturally evolved transportation properties of these biogenerated vesicles can lead to precise cellular/intracellular targeting and drug release, without triggering side effects or adverse immunological reactions. A prominent example of such carriers are exosomes, which are endosome-derived microvesicles released by mammalian cells and found in simple body fluids [74]. They have a complex architecture, composed of lipids, such as phosphatidylserine (PS), cholesterol, sphingomyelin (SM), prostaglandins, and fatty acids such as arachidonic acid, which provide them with structural integrity and stability, along with certain proteins, genetic components (DNA/RNA), and other biomolecules. Although still in its infancy, several literature reports have already exemplified the use of exosomes, derived from blood, saliva, cerebrospinal fluids, etc., as specific disease biomarkers, and carriers of active agents such as RNA for therapeutic purposes. Another class of biologically derived lipid-based carriers is outer membrane vesicles (OMVs), which are small spherical proteoliposomes derived from the cell wall of Gram-negative bacterial cells. OMVs have evolved as promising nanocarriers for vaccine development and drug delivery [75]. Being of natural origin, these drug carriers are readily accepted in the body, can effectively evade the immunosurveillance system, freely travel across various biological barriers, bind with specific host cells, etc.

The various types of synthetic lipid-based nanocarriers are listed in Figure 2.

## 4. Lipid-Based Inhalable Delivery of Active Agents for Treating NSCLC

The various kinds of lipid-based micro- and nanocarriers, as listed in the previous section, have been extensively used for the inhalable delivery of active agents (chemotherapy drugs, therapeutic genes, peptides, etc.) in the lungs for the treatment of NSCLC, representative examples of which are provided in this section. These lipid-based formulations have shown promising results in in vitro and in vivo studies, and several of them are undergoing clinical trials, such as: phase I: a cholesterol–fus1 liposome complex; phase II: a BLP25 liposome vaccine, liposomal lurtotecan; phase III: an irinotecan liposome injection, lipoplatin; and phase IV: a paclitaxel liposome [76].

### 4.1. Lipid Nanoparticles for Delivery of Chemotherapy Drugs

The inhalable delivery of therapeutic agents attached to liposomes can be achieved either using a nebulizer or dry powders inhalers. The aerosolization via nebulization may lead to structural disintegration which compromises the efficiency of the drug. Hence, the dry powder option is preferred as these formulations are stable and can be produced by spray-drying, spray freeze drying, or freeze drying, which is then followed by micronization [77]. Zhang et al., formulated a liposomal dry powder inhaler loaded with curcumin (LCD), which was employed for treating primary lung cancer (Figure 3). Curcumin has anticancer properties but finds limited usage due to its poor hydrophilicity and bioavailability, and it is also rapidly cleared from the body. Liposomes were prepared from soya bean lecithin and cholesterol using the film hydration method. Then, mannitol was added to the liposomes and the mixture was kept in a lyophilizer for freeze-drying. The LCDs obtained were of spherical morphology and monodispersed, with an average diameter of 94.65 ± 22.01 nm. The LCDs performed better than free curcumin powder upon pulmonary inhalation, as shown by better lung deposition. Moreover, the LCDs had finer particle fractions and aerodynamic diameters. Curcumin incorporated within the liposomes showed a better selection index than free curcumin. The formulation was highly cytotoxic towards the A549 cell line but did not exhibit any toxicity towards normal bronchial epithelial cells (BEAS-2B). The in vivo studies were also conducted using Sprague Dawley rats with orthotopically-implanted lung cancer. Their lungs were sprayed through the trachea with three formulations: (a) curcumin attached with liposomal powder, (b) curcumin powder, and (c) gemcitabine. The liposomal formulation performed better than the other two; therefore, the formulation can be improved further and used for the intratracheal treatment of NSCLC. Figure 3 shows: TEM images of liposomal curcumin; a comparison of cell viabilities in the case of curcumin powders, liposomal curcumin powders, and gemcitabine; confocal laser scanning microscope (CLSM) images of A549 cells incubated with curcumin powders and liposomal curcumin powders; the lungs’ appearance; sections of lung tissue; VEGF expression for the lung cancer model in rats and lung cancer rats treated with liposomal curcumin powder; and the effect of different formulations on MDA (oxidation indicator) [78].

Xu et al., used vincristine and incorporated it into liposomes. It was needed to increase lung exposure and decrease the clearance time, in order to improve the efficiency of vincristine. This was achieved by making use of pulmonary delivery and spray-drying techniques to yield a formulation that can be administered via dry powder inhalers. The liposomal formulation ensured the sustained release of the drug. In vitro studies established better antitumor performance of the drug when incorporated in liposomes, as compared to its free form. The pharmacokinetics were also substantially better in the liposomal formulation, with an enhanced accumulated concentration, systemic exposure, and a reduced elimination time [79].

9-Nitrocamptothecin (a derivative of camptothecin) is an anticancer drug that does not dissolve in water. Knight et al., incorporated the drug in a liposomal formulation and used it to treat murine melanoma and human osteosarcoma pulmonary metastases in mice models. The formulation was made using dilauroylphosphatidylcholine, and the nebulized particles were of the size range of 1.2–1.6 m. The formulation in aerosol form was administered to mice for a duration of 15 min to 2 h every day. The growth of the subcutaneous tumor was significantly reduced only after a few weeks of treatment. However, when the same formulation was administered orally, no significant cytotoxic effect against the tumor was observed. This confirmed that the efficacy was due to pulmonary deposition. The activity of the formulation when administered intramuscularly was also significantly less than the inhalable aerosol. The efficacy of the above formulation in inhalable form endowed it with the potential to be used for the treatment of pulmonary metastases [80]. In a similar study, camptothecin was incorporated into a liposomal formulation using diauroylphosphatidylcholine. The pharmacokinetics and therapeutic potential were explored using a mouse model. Only after 30 min of inhalation, a substantial amount of drug was accumulated in the lungs, while the accumulation in other organs was minimal. In contrast, the accumulation in the lungs was not appreciable when administered intramuscularly [81].

In another study, a liposomal formulation of pirfenidone was employed against NSCLC. Pirfenidone is a known antifibrotic agent which was approved almost a decade ago for the treatment of idiopathic pulmonary fibrosis. The drug was passively loaded on the liposomal formulation, which was fabricated using the thin-layer hydration technique. The TEM data revealed spherical morphology with uniform distribution and an average diameter of 211.8 nm. The formulation had appreciable aerosolization properties. The MTT assay results established that the liposomal formulation was more cytotoxic toward A549 cells as compared to the free drug. Further in vitro studies carried out using human embryonic kidney cells (HEK-293) showed no toxicity with the formulation, which establishes the selectivity of the liposomal formulation towards cancer cells [82]. Ghosh et al., co-loaded vincristine and doxorubicin onto a PEGylated liposomal formulation and studied its performance against NSCLC. Loading was carried out against a modified ammonium ion gradient, and the encapsulation was about 95% for both drugs. The formulation exhibited a unilamellar-spherical structure with an average size of about 94 nm. The dual drug formulation showed even better cellular uptake than liposomal doxorubicin. The cell viability of A549 cells was greatly reduced and there was enhanced tumor regression as compared to the single drug formulation of liposomal doxorubicin. The in vivo studies also corroborated the enhanced efficiency of the co-encapsulated PEGylated liposomal formulation [83].

Paclitaxel is another common anticancer drug that can be used for the treatment of patients suffering from lung cancer. The drug, however, is quite lipophilic in nature, and it becomes a challenge to solubilize it in an aqueous medium, which is essential for oral and intravenous administration. Liposomes can overcome this challenge and can improve pharmacological properties and reduce toxicity. A liposomal formulation loaded with paclitaxel was prepared in the presence of butanol along with dilauroylphosphatidyl choline. The drug-to-lipid ratio (weight/weight) was fixed at 1:10, and then the system was lyophilized at a very low temperature (−70 °C) to yield the desired liposomal formulation. The aerosolized particles were later characterized with the help of an Andersen cascade impactor; the particles exhibited a mass median diameter of value equal to 2.2 m. Pharmacokinetic studies were carried out and it was observed that the area under the curve was 26 times higher as compared to the case of intravenous injection. The in vivo studies were carried out using BALB/c mice, which were inoculated with tumor cells and divided into three groups for further evaluation. One group was left untreated (control) (group 1), one was treated with blank liposomes (group 2), and the third group was intratracheally treated with liposomes loaded with paclitaxel (group 3). The mice of group 3 showed a lower lung weight as compared to the mice of the other two groups, which suggested appreciable tumor reduction. The inhalation of the paclitaxel-loaded liposomal formulation also showed better long-term survival in the diseased mice [84].

Adel et al., utilized proliposomes to deliver curcumin to the lungs. Proliposomes are a modified form of liposomes that are entirely dry and free-flowing powders made out of lipid vesicles. The drug is encapsulated inside the bilayer structure. These were prepared via nano-spray drying making use of hydroxypropyl beta-cyclodextrin as carriers, lecithin, and cholesterol as lipids, stearyl amine as a positive charge inducer, and Poloxamer 188 as a surfactant. The formulation showed excellent aerosolization and could accumulate in the deep lung tissues with a high fine particle fraction. The formulation showed enhanced cytotoxicity against A549 cells along with significant downregulation of proinflammatory cytokines, as compared to the case of pure drugs. The formulation was better than the free drug in the extent and rate of absorption by the lungs and the retention time as confirmed by the pharmacokinetics analysis [85]. Gaspar et al., fabricated PEGylated liposomes conjugated with transferrin (Tf) and then loaded them with doxorubicin. The formulation was delivered to an orthotopic lung cancer model in athymic Rowett nude rats with the help of an intracorporeal nebulizing catheter. The rats were given different doses and formulations of doxorubicin and their performance was compared to the liposomal system. The animals did not survive for long upon intravenous administration. The survival of animals was better in the case of the Tf-conjugated liposomal formulation delivered via inhalation [86].

In a very recent and interesting study, a live carrier, bacteria, was used to deliver a liposomal formulation containing paclitaxel, and its efficiency was explored in the inhalable treatment of lung cancer (Figure 4). The liposomal formulation of paclitaxel was efficiently internalized inside the bacteria (*E. coli* and *L. casei*) via the process of electroporation [87]. The loading of the formulation had no apparent effect on the growth phase of the bacteria. The formulation-loaded bacteria performed better than the simple mixture of the formulation and bacteria, and it significantly reduced the proliferation of A549 cells. The liposomal formulation encapsulated in *E. coli* exhibited maximum efficiency against rat lung cancer in vivo along with the suppression of HIF-1 and VEGF and the enhancement of apoptosis. In addition, the expression of immune cells and markers was enhanced. The bacterial formulation showed very high accumulation in the lungs as compared to other organs after intratracheal administration, making it a promising candidate. Figure 4 depicts a TEM image of liposomal paclitaxel and bacteria *E. coli* and *L. casei* upon electroporation to encapsulate the liposomal formulation; CLSM images of A549 cells incubated with formulation-loaded bacteria; the colony amounts of the formulation-loaded bacteria in different organs after inhalation; comparison of leukocytes in the blood in the case of a healthy mouse, a mouse with lung cancer, and a mouse with lung cancer treated with different formulations.

In a study conducted by Nassimi et al., detailed toxicological profiling of SLNs was completed to evaluate their safety in the biological system. The study established the therapeutic window for SLNs and the toxic dosage with the help of A549 cells and murine precision-cut lung slices. In vivo studies were also performed using female BALB/c mice which were exposed to different amounts of SLNs every day. The toxic dosage for in vitro, ex vivo, and in vivo models were evaluated. Carbon black was employed as a control particle, and at corresponding concentrations, it caused cytotoxic and inflammatory effects. The findings help to confirm the optimal concentration of SLNs for the murine inhalation model [88]. Bakhtiary et al., reported the development of an erlotinib-loaded SLN formulation in the form of a dry powder inhaler. The formulation was made using a fixed amount of Compritol/poloxamer 407. The formulation was spherical with a size of less than 100 nm with good encapsulation efficiency. The cytotoxicity of the cargo drug was enhanced when encapsulated in liposomes, as confirmed by an MTT assay and DAPI staining using A549 cells. The formulated dry powder is composed of microparticles with an optimum flow and aerodynamic properties. Deep inhalation of the formulation was also established, which makes the formulation a potential treatment option for NSCLC [89]. In a similar study, Nafee and co-workers incorporated myricetin in surfactant-free SLNs. The formulation was characterized by several key features and parameters, physicochemical properties, drug loading, and release profile, cellular uptake, and anticancer activity. The formulation was converted into inhalable microparticles by spray-drying them along with carbohydrate carriers, with good flowability. The cytotoxicity was also enhanced, which could be attributed to higher cellular uptake. The higher uptake was confirmed using confocal imaging and doubled fluorescence sensing [90].

In another study, NLCs were synthesized and optimized with the help of a data analysis model with a Box–Behnken design. The NLCs were prepared using an emulsification technique and loaded with paclitaxel. The Box–Behnken design helped to optimize the formulation and decipher the effect of various processing variables on the characteristics of NLCs. The model suggested that the most applicable factors for maintaining the size and monodispersity were a higher surfactant content, a lower lipid ratio, and a moderate homogenization speed. The different formulations exhibited significant cellular uptake, and spray-dried NLCs showed good flow properties, which can be used for delivering drugs to deeper airways. The drug was effectively localized in the lungs via pulmonary delivery, as shown by in vivo experiments. In addition, the surfactants used were beneficial in overcoming drug resistance caused by P-gp efflux [91]. Patel et al., evaluated the anticancer activity of celecoxib incorporated in NLCs using an in vivo model, as a single treatment mode and in combination with docetaxel administered intravenously. The formulation showed significant anticancer activity, which was even higher when the combination therapy was used [92].

Patil et al., incorporated bedaquiline, an FDA-approved drug with proven efficacy against NSCLC, within lipid-based nanocubosomes using a solvent evaporation technique. This drug-loaded nanoformulation showed excellent aerodynamic properties for inhalable delivery following nebulization. Treatment of this formulation on A549 lung cancer cells showed superior cellular internalization, anti-proliferation, inhibited colony formation, and suppressed metastasis in vitro when compared to that obtained using the free drug. Higher antitumor activity was also demonstrated in a simulated 3D in vivo model for the drug-loaded nanoformulation as compared to that of the free drug [93].

### 4.2. Lipid Nanoparticles for Gene Delivery to the Lungs

Genetic materials, such as DNA and RNA, have long been perceived as viable therapeutic agents against a variety of diseases, including cancer [94]. Since viruses are natural gene carriers, various genetically engineered viruses containing functional genes were first employed for gene therapy. However, several clinical complications, including mutagenesis and immunogenic shock, dampened the enthusiasm related to ‘viral vectors’ [95]. This led to the development of synthetic, micro/nanoparticulate gene carriers as ‘non-viral vectors’ as an alternative to engineered viruses [96]. In this context, lipid-based nanocarriers of genetic components have emerged as highly promising candidates owing to their stability, high amount of gene loading, protection of genes from physiological degradation, and delivery to target cells with high efficacy and less toxicity [97].

In the context of gene therapy in cancer, three types of genetic materials can be primarily used [98]. The first type is plasmid DNA that includes tumor-suppressor genes such as p53. These are engineered double-stranded DNA of about 2–6 kilo base pairs (kbps) in length. The other type is short interfering RNA (siRNA), which are double-stranded RNA of 24 kbps in length, that blocks the expression of tumor-promoting oncogenes such as bcl-2. The third type is messenger RNA (mRNA), which encodes for proteins that enhance the body’s general immunity against cancer cells/tissues. Such therapeutic mRNA forms the basis for the development of cancer vaccines, which can either treat pre-existing cancer via cancer immunotherapy or act as a prophylactic by defending the body against potential cancer development, particularly those caused by viruses such as human papillomavirus (HPV). In each case, the genes are negatively charged and thus require moderately cationic carriers to form a stable electrostatic complex. Cationic lipids, in combination with anionic/neutral lipids in an optimal ratio, are ideal for the development of micro/nanocomplexes for gene delivery [99]. These gene micro/nanocomplexes are either used alone or in combination with moderate dosages of chemotherapeutic drugs, often co-incorporated within the same carrier, for therapeutic purposes against cancer.

In one of the early reports, Choi et al., used cationic solid lipid nanoparticles (SLNs), comprised tricaprin (TC), 3β[N-(N′, N′-dimethylaminoethane) carbamoyl] cholesterol (DC-Chol), dioleoylphosphatidylethanolamine (DOPE), and Tween 80 for the complexation of a plasmid DNA encoding both the marker enhanced green fluorescent protein (EGFP) gene and the functional p53 gene. The (pp53-EGFP)/SLNs complexes were formed using a modified melt homogenization method. Robust gene transfection efficiency in vitro, with higher efficiency than that shown by the commercial transfection agent Lipofectin^®^, was established in human NSCLC (H1299) cells with the (pp53-EGFP)/SLNs complexes. As mutation in the p53 gene is a leading genetic cause of lung cancer, the restoration of wild-type p53 function following successful gene transfer with the SLNs rescued the apoptotic pathway in the cancer cells and impeded their proliferation [100].

Hybrid nanoformulations (HNF) of lipids can also be beneficial for the treatment of NSCLC. In one study, hybrid lipid nanoformulations were made using poly(lactic-co-glycolic) acid and dipalmitoyl phosphatidyl choline to deliver siRNA. siRNAs are known to suppress the genes which are responsible for the pathogenesis of severe lung conditions. The HNF showed good physicochemical properties, and triphasic release of siRNA was observed over 5 days. The formulation was delivered using a vibrating mesh nebulizer and the aerosol performance was optimal for in vitro experiments. X-ray scattering analysis was performed to depict the exceptional stability of the formulation upon incubation with artificial mucus, confirming that it could be used for aerosolized delivery in the mucus-lined respiratory pathway. Experiments performed using a triple cell co-culture model confirmed that the formulation was effectively internalized by the airway epithelia cells without any apparent cytotoxicity or inflammation. In A549 cells, sustained inhibition of the sodium transepithelial channel was observed when siRNAs were delivered with the help of the formulation [101]. Some other lipid formulations for inhalable delivery for treating NSCLC are discussed in Table 2 below.

Garbuzenko et al., fabricated multifunctional NLCs for the simultaneous delivery of paclitaxel and siRNA. The drug would induce cell death due to its anticancer properties and siRNAs would suppress the four types of EGFR-TKs (epidermal growth factor receptors–tyrosine kinase). The passive targeting was promoted by inhalable administration and the formulation was given active targeting ability by using a decapeptide (luteinizing hormone–release hormone). The in vitro studies were carried out using A549 and in vivo by using the orthotopic NSCLC mouse model. The formulation exhibited desirable organ accumulation and superior anticancer activity as compared to the single drug or siRNAs or non-targeted delivery. The multifunctionality will also reduce the adverse effects of the treatment [102].

In a recent article, Kim et al., demonstrated that optimized PEG content in lipid-based nanocarriers can help in nebulizer-based delivery by providing higher structural stability against shear stress and effective traversal through the mucosal barrier. By further incorporating the cholesterol analog, β-sitosterol, within these lipid-nanocarriers, polyhedral geometry and improved endosomal escape were achieved. In a mouse-model, the authors demonstrated high transfection efficiency and sustained protein production following the delivery of mRNA incorporated within this nanocarrier following inhalation delivery. Although the authors showed the functional effect of this mRNA delivery technique against cystic fibrosis, similar formulations can be developed for the treatment of NSCLC [103].

The quality by design (QbD) approach employs statistical, analytical, and risk-management models to optimize the design and development of medicinal formulations. Bardoliwala et al., reported the use of the QbD approach to systematically develop and characterize an inhalable dry powder formulation of a hybrid lipid–polymer nanocarrier, co-loaded with the chemotherapeutic docetaxel and the multidrug-resistance overcoming shRNA targeting the p-glycoprotein gene (ABCB1 shRNA). Although no biological data against lung cancer have been reported, the authors could optimize the particle size, aerodynamic properties, drug and shRNA loading, and high particle fraction ideal for DPI development [104].

**Table 2 pharmaceutics-15-01457-t002:** Some lipid-based nanoformulations for inhalable delivery to treat NSCLC.

S. No.	Lipid Formulation and Composition	Therapeutic Agent	Salient Feature	Ref.
1	Liposome: (a) dipalmitoyl phosphatidylcholine, (b) dipalmitoyl phosphatidylglycerol, and (c) dipalmitoyl phosphatidic acid	5-Fluorouracil	The toxicity to healthy cells was reduced and the formulation provided sustained release of the drug in the lungs, which reduced the frequency of drug administration.	[105]
2	Liposomes: cholesterol and 16:0 1,2-dipalmitoyl-sn-glycero-3-phosphocholine (DPPC)	Indomethacin	The formulation was tested on A549, H1299, and H460 cell lines. The formulation showed optimum physicochemical properties and improved efficacy in in vitro and ex vivo studies.	[106]
3	Liposomes: cholesterol and DPPC	Osimertinib	The formulations (active and passive) exhibited great aerosolization properties. The formulation showed enhanced cytotoxicity against NSCLC and inhibited tumor cell migration and colonization, as shown by in vitro assays.	[107]
4	Nano-emulsion (o/w)	Naringin and celecoxib	The formulation exhibited dose-dependent cytotoxicity, and the toxicity was greater than the combination of these drugs in a solvent.	[108]
5	Nano-emulsion: lauric fatty acids, palm kernel oil ester (medium-chain fatty acids), myristic fatty acids (long-chain), and lecithin.	Docetaxel	The formulation was made using the high-energy emulsifying technique. The formulation inhibited the growth of A549 cells and did not cause any noticeable cytotoxicity to normal cells.	[109]
6	Nano-emulsion: super refined L 18 POG	Erlotinib	The formulation exhibited enhanced efficiency as observed during in vivo and ex vivo studies.	[110]
7	SLNs: soy lecithin, Compritol 888 ATO, and Poloxamer 188.	Epirubicin	The formulation suffered minimum drug loss and could efficiently achieve deep lung delivery. In vivo studies showed that the formulation was more efficacious than the free drug. Similar findings were obtained using male Sprague Dawley rats.	[111]
8	SLNs	Paclitaxel and curcumin	The formulation gave a better therapeutic effect than the simple combination of drugs against A549 cells. The tumor was also ablated efficiently in the mice model due to the synergistic effect of the drugs.	[112]
9	NLCs: Precirol ATO 5 (solid lipid), squalene (liquid lipid), and soybean phosphatidylcholine	Paclitaxel	The formulation exhibited higher cytotoxicity against A549 cells as compared to gemcitabine. The formulation had significant accumulation and retention in the lungs of the orthotopic NSCLC mice via inhalation. No apparent toxicity was observed in the major organs.	[113]
10	NLCs: Cremophor EL	Paclitaxel and doxorubicin	Organ distribution studies were performed which confirmed the high drug distribution as compared to other formulations and free drugs. Animals treated with the DPI of the formulation showed no apparent signs of tissue damage.	[114]
11	NLCs: Stearic acid and phosphatidyl choline	9-Bromo-noscapine	The formulation exhibited enhanced cellular uptake and apoptosis as compared to the free drug.	[115]
12	NLCs: DSPE-PEG-2000-PE, DSPC, DSPE-PEG-NHS, and DOTAP	Paclitaxel, Gefitinib, and siRNA	The complex drug delivery system with multiple components showed higher activity against NSCLC as compared to any individual component.	[116]

### 4.3. Other Therapies

Several other therapeutic regimens, such as immunotherapy, photoactivated therapies, sonodynamic therapy (SDT), etc., are being actively explored as experimental therapeutics in lung cancer [117,118]. Cancer immunotherapy is being actively pursued in several cancers, including lung cancer, as it not only affects the primary tumor but also seeks and destroys metastatic cells. Typically, since cancer immunotherapy is a systemic rather than a localized phenomenon, immunomodulatory agents are delivered either intravenously or intraperitoneally. However, some instances of localized, pulmonary delivery of immunotherapy drugs can be found, which are being developed as ‘nasal vaccines’ against lung cancer. For example, Loira-Pastoriza et al., formulated a liposomal formulation incorporating the immunomodulatory CpG oligonucleotide. Direct, pulmonary delivery of the CpG incorporated liposomes in a murine model of B16F10 metastatic lung cancer resulted in delayed tumor growth and impeded metastatic spread. The immunomodulatory mechanism was evident from the enhanced production of pro-apoptotic proteins, and IFNγ, MIG, and RANTES, T helper type 1 cytokines and chemokines, in the lung [119].

Xiao et al., explored the treatment of metastatic lung cancer by employing a combination therapy comprising chemotherapy and SDT (Figure 5). The concept of combination therapy or multimodal therapy often enhances efficacy due to synergism and at the same time reduces the dosage which in turn would reduce the adverse effects. In their study, they worked with cationic liposomal hydroxycamptothecin (CLH) and 5-aminolevulinic acid. The CLH was fabricated out of soybean lecithin, octadecylamine, and cholesterol using the thin film method. Upon completing in vitro studies, it was established that Chemo-SDT was more cytotoxic towards tumor cells as compared to only SDT and only using cationic liposomal CLH (intratracheally or intravenously). In vivo studies were also carried out using mice with metastatic lung tumors; the experimentation showed that combined chemo-SDT showed inhaled chemo-SDT therapy was most effective and the synergism of the two modes of treatment significantly enhanced the efficacy. Figure 5 shows that CLH was obtained from hydroxycamptothecin (along with a lipid mixture), and the active metabolite of 5-ALA is Protoporphyrin IX which will work as a sonosensitizer, and images for the expression of terminal deoxynucleotidyl transferase dUTP nick end labeling (Tunel) and hematoxylin and eosin (H &E) staining of lung tissues in the case of the mouse model with lung cancer and the mouse model with lung cancer treated with CLH (intravenously and intratracheally) only, SDT only, and chemo-SDT [120].

## 5. Challenges and Opportunities

It is now well established that the direct, inhalable route is more advantageous than systemically administered drug delivery to the lungs. Advances in inhalable drug-delivery techniques, such as nebulization, pMDI, DPI, etc., have further made this route efficacious. Already, several clinically acceptable devices and formulations for inhalable drug delivery for the treatment of a myriad of diseases, such as asthma, cystic fibrosis, COPD, lung cancer, etc., are available. Yet, a number of hurdles still remain that need to be overcome for the routine clinical application of lipid-carrier-based inhalable drug delivery to the lungs. These challenges also present several exciting opportunities in this area, both from the point of view of basic research and clinical development.

The first challenge is to design a stable lipid-based drug nanoformulation that is structurally robust to withstand physical factors such as ultracentrifugation, freeze drying, the shear force of nebulizers and propellants, etc. Research has demonstrated that the structural integrity of lipid-based formulations can be well preserved with the incorporation of cholesterol, its analogs, polymers such as polyethylene glycol (PEG), etc., within the structure/surface of the micro/nanocarriers. With regards to the loading of therapeutic payloads, lipid-based nanoparticles can be suitably designed to incorporate high amounts of both hydrophilic and lipophilic drugs. Anionic genetic materials, such as DNA and RNA, can be effectively complexed using controlled amounts of cationic lipids as components of the carriers. The next challenge is to safely traverse across the mucosal barrier and avoid being captured by the pulmonary macrophages. Nano/micro carriers with a hydrophilic surface and neutral/mildly anionic surface charge are ideally suited to overcome this challenge. Since lipid-based carriers are primarily lipophilic, surface passivation with inert polymers such as PEG or dextran is critical in designing such smart carriers. The effect of size (in the nano/micron range) for deposition in various lung regions has already been discussed earlier in this review. Although a large body of literature is already available regarding the ideal design parameters for effective inhalable delivery, more research is needed, involving both wet-chemistry and computational models, for further optimization of these parameters.

Being a direct method of drug delivery, inhalable carriers require a much lesser drug dosage than systemic drug carriers; moreover, the possibilities of off-target effects are minimal in direct drug delivery. Therefore, drug-carrying inhalable micro/nanoformulations comprised biocompatible lipids pose little toxicological threat in treated animals/patients. However, very small nanoparticles (diameter below 20 nm) can infiltrate the deep lungs and further permeate into the systemic circulation, from where they can reach other, non-target organs. Thus, controlling the dimensions of the carriers is critical in not only delivering optimal amounts of therapeutic agents to the desired lung sites but also controlling their long-term persistence and possible deleterious effects on the body.

As evident from the examples presented in Section 4, although several types of lipid-based nano- and microcarriers exist, most of the reported work so far has used liposomal- and SLN-based formulations for inhalable delivery against lung cancer. This provides opportunities for exploring other types of lipid-based nanocarriers for the same purposes, especially those with natural origins (e.g., exosomes). In addition, there are possibilities for devising other therapeutic approaches against NSCLC, such as immunotherapy, photodynamic therapy (PDT), etc., involving lipid-based nanoparticles and inhalable delivery. Combination therapy approaches involving such nanoformulations should also be explored against NSCLC. Image contrast agents, such as the MRI probe gadolinium, can also be incorporated with these carriers for theranostics and image-guided therapy.

## 6. Conclusions

It is amply evident that the inhalable delivery of micro- and nanoparticulate systems incorporating one or multiple therapeutic agents is highly promising for the treatment of NSCLC. The various drawbacks associated with the currently administered systemic delivery of chemotherapeutics, such as poor tumor accumulation, drug loss, off-target effects, systemic and organ toxicity, etc., can be effectively avoided using inhalable delivery formulations. The ease and user-friendliness of this delivery mode can pave the way for effective treatment in non-clinical and remote settings. Moreover, by effective manipulation of the size, surface properties, and aerodynamic features of these micro/nanocarriers, they can be targeted to deeper lung regions (the alveoli) and can also intravasate to the systemic circulation by traversing the air–blood barrier. Besides chemotherapy, a number of experimental therapies, such as gene therapy, immunotherapy, photoactivated therapies, etc., are being actively explored in the treatment of NSCLC and other lung diseases using inhalable micro/nanocarrier formulations. Following the global devastation caused by various microbial diseases in the recent past, the production of inhalable, nasal vaccines has witnessed unprecedented developments. Overall, in the near future, we anticipate several exciting developments in the realm of inhalable drug delivery for the treatment of NSCLC and several other diseases.

## Figures and Tables

**Figure 2 pharmaceutics-15-01457-f002:**
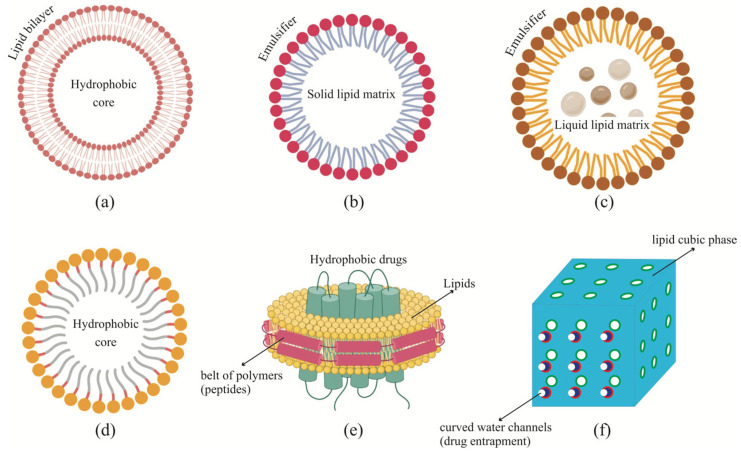
Common synthetic lipid-based drug carriers: (**a**) liposomes, (**b**) SLNs, (**c**) NLCs, (**d**) micelles, (**e**) nanodiscs, and (**f**) cubosomes.

**Figure 3 pharmaceutics-15-01457-f003:**
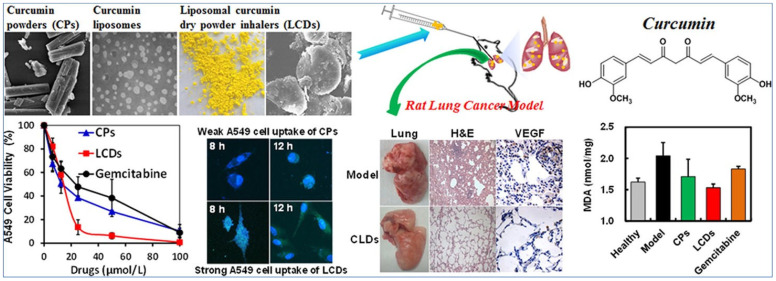
Inhalable treatment for lung cancer, in vitro and in vivo studies, using curcumin-loaded liposomes [78], copyright (2018) @ Elsevier.

**Figure 4 pharmaceutics-15-01457-f004:**
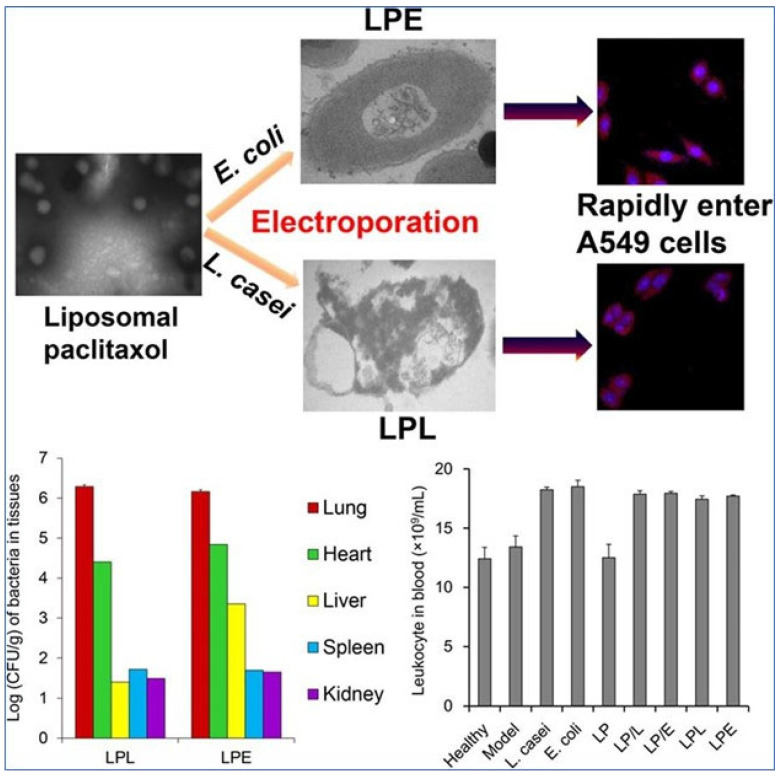
Schematic representation of the liposomal delivery of paclitaxel with the help of bacteria for in vitro and in vivo studies [87], copyright (2020) @ Elsevier.

**Figure 5 pharmaceutics-15-01457-f005:**
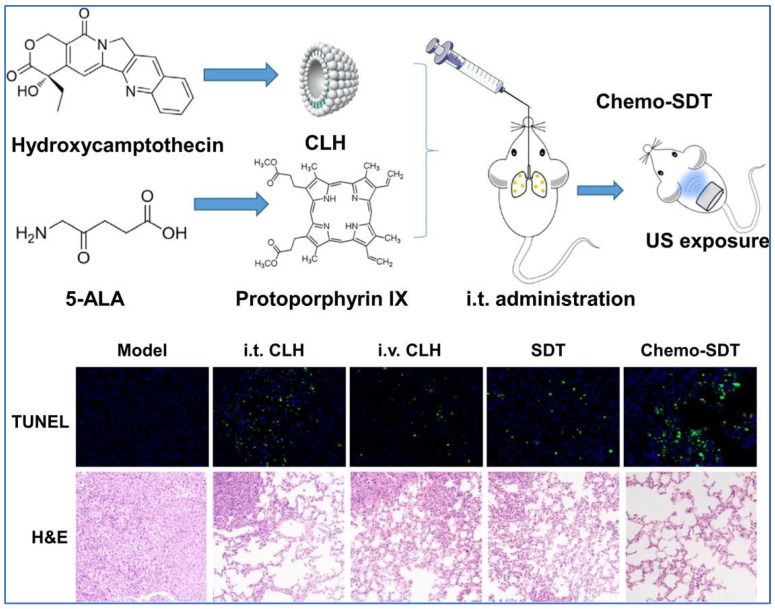
Schematic representation of chemo-sonodynamic therapy [120], copyright (2021) @ Elsevier.

**Table 1 pharmaceutics-15-01457-t001:** FDA-approved lipid-based drug formulations.

S. No.	Trade Name	Manufacturer	Year of Approval	Usage
1	Spikevax	Moderna	2022	COVID-19 vaccine, mRNA
2	Comirnaty	Pfizer and BioNTech	2020	COVID-19 vaccine, mRNA
3	Lipusu	Luye Pharmaceuticals	2020	Squamous NSCLC and esophageal cancer
4	Onpattro	Alnylam Pharmaceuticals	2018	Transthyretin-mediated amyloidosis
5	Shingrix	GlaxoSmithKline Biologicals	2018	Shingles and post-herpetic neuralgia
6	Arikayce	Insmed Inc.	2018	Lung disease
7	Vyxeos	Jazz Pharmaceuticals	2017	Acute myeloid leukemia
8	Onivyde	Ipsen	2015	Metastatic pancreatic cancer
9	Thermodox	Celsion Corporation	2014	Hepatocellular carcinoma
10	Ikervis	Santen Pharmaceutical Co.	2014	Keratitis
11	Marqibo	Acrotech Biopharma	2012	Acute lymphoblastic leukemia
12	Exparel	Pacira Pharmaceuticals, Inc.	2011	Pain management
13	Octocogalfa (Advate)	Bayer Pharma AG	2009	Hemophilia A
14	Depodur	SkyPharma Inc.	2004	Pain management
15	Mepact	Takeda Pharmaceutical Limited	2004	High-grade, resectable, non-metastatic osteosarcoma
16	Estrasorb	Novavax	2003	Menopause therapy
17	Visudyne	Bausch and Lomb	2000	Wet age-related macular degeneration, myopia, and ocular histoplasmosis
18	Myocet	Elan Pharmaceuticals	2000	Metastatic breast cancer
19	AmBisome	Gilead Sciences	1997	Fungal/protozoal infections
20	DaunoXome	Galen	1996	Kaposi’s sarcoma
21	Doxil	Janssen	1995	Kaposi’s sarcoma, ovarian cancer, and multiple myeloma

## Data Availability

This review article is based on the existing literature and no new data were created.

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
