# Peer review of "Lipid-Based Inhalable Micro- and Nanocarriers of Active Agents for Treating Non-Small-Cell Lung Cancer"

_pharmaceutics, 2023, doi:10.3390/pharmaceutics15051457_

Round 1

Reviewer 1 Report

The manuscript “Lipid-based inhalable micro- and nanocarriers of active agents for treating Non-small Cell Lung Cancer” seems interesting and informative. However, the following points should be considered before publication.

1.   Background for inhalable drug delivery is not sufficient. Discuss more on inhalable/pulmonary drug delivery systems.

2.   Discuss key parameters to be considered for designing lipid-based inhalable micro/nanoparticles.

3. Which is better micro or nano-sized nanocarriers for pulmonary drug delivery? Discuss more on this.

4.  Discuss the limitations/challenges of existing strategies in inhalable drug delivery.

5. Discuss future perspectives and research direction.

Author Response

Referee 1:

The manuscript “Lipid-based inhalable micro- and nanocarriers of active agents for treating Non-small Cell Lung Cancer” seems interesting and informative. However, the following points should be considered before publication.

  1. Background for inhalable drug delivery is not sufficient. Discuss more on inhalable/pulmonary drug delivery systems.

Authors: We thank the reviewer for the critical comments. We have updated the background section as suggested.

  1. Discuss key parameters to be considered for designing lipid-based inhalable micro/nanoparticles.

Authors: We thank the reviewer for their valuable comment. We have discussed some important parameters that should be considered for inhalable delivery, towards the end of Introduction section of the revised manuscript.

  1. Which is better micro or nano-sized nanocarriers for pulmonary drug delivery? Discuss more on this.

Authors: We have discussed the effect of size (in both nano- and micron-range) on inhalable delivery in this revised manuscript.

  1. Discuss the limitations/challenges of existing strategies in inhalable drug delivery.

Authors: We are thankful to the reviewer for this very important comment, and have introduced a new section entitled “Challenges and Opportunities” in this revised manuscript, which effectively addressed this issue.

  1. Discuss future perspectives and research direction.

Authors: We have introduced a new section entitled “Challenges and Opportunities” in this revised manuscript, which effectively addressed this issue.

Reviewer 2 Report

Indrajit Roy submitted an interesting review about the lipid-based nano/micro carriers for NSCLC therapy. Various types of carriers were discussed, and the application thereof was mentioned. The review fell within the scope of Pharmaceutics, and might be considered for publication after Major Revision. Detailed comments:

1. For the inhalable drug delivery types, in addition to nebulizer and DPI, MDI is also an important category. Please make proper introduction about MDI.

2. Some products were listed in Table 1. However, some new products had been approved recently after the COVID-19 outbreak. Please supplement these products.

3. Besides of the lipid-based systems introduced in Section 2, exosomes and outer membrane vesicles should also be mentioned.

4. Please add the copyright access courtesy statement in Fig. 3~5, etc.

5. The contents in the later sections were not in parallel with the Introduction Section. For instance, nebulizer and DPI were mentioned in the Introduction Section, but the following part did not comprehensively summarize the application of nanoparticle based nebulizer or DPI. It was suggested to add the corresponding information.

6. Before Conclusion Section, it was advised to add a part of discussion to offer the authors’ expert opinion on this field.

7. According to the Reference list, quite few papers published in recent 3 years (2021~2023) were cited. Please cite more recent papers, if possible.

Author Response

Referee 2:

Indrajit Roy submitted an interesting review about the lipid-based nano/micro carriers for NSCLC therapy. Various types of carriers were discussed, and the application thereof was mentioned. The review fell within the scope of Pharmaceutics, and might be considered for publication after Major Revision. Detailed comments:

  1. For the inhalable drug delivery types, in addition to nebulizer and DPI, MDI is also an important category. Please make proper introduction about MDI.

Authors: We have included MDI in the introductory discussions

  1. Some products were listed in Table 1. However, some new products had been approved recently after the COVID-19 outbreak. Please supplement these products.

Authors: We have added recently approved products in the table, as per reviewer’s suggestion. We have expanded the table and also added few more noticeable products approved in the last decade.

  1. Besides of the lipid-based systems introduced in Section 2, exosomes and outer membrane vesicles should also be mentioned.

Authors: We have briefly discussed about these two types of lipid-based natural drug carriers in this revised manuscript

  1. Please add the copyright access courtesy statement in Fig. 3~5, etc.

Authors: We apologise for not adding it in the manuscript. We have taken care of this in the revised manuscript.

  1. The contents in the later sections were not in parallel with the Introduction Section. For instance, nebulizer and DPI were mentioned in the Introduction Section, but the following part did not comprehensively summarize the application of nanoparticle based nebulizer or DPI. It was suggested to add the corresponding information.

Authors: We have tried to add the required information to the best of our ability.

  1. Before Conclusion Section, it was advised to add a part of discussion to offer the authors’ expert opinion on this field.

Authors: We are thankful to the reviewer for this very important comment, and have introduced a new section entitled “Challenges and Opportunities” in this revised manuscript, which effectively addressed this issue.

  1. According to the Reference list, quite few papers published in recent 3 years (2021~2023) were cited. Please cite more recent papers, if possible.

Authors: We have added more recent references in the paper, as per the learned referee’s comment. The added references are cited in the revised manuscript in Table 2.

Reviewer 3 Report

This manuscript presents a valuable review of the use of lipid-based nanoparticles as inhalable therapeutics for non-small cell lung cancer (NSCLC) treatment. However, several issues need to be addressed to enhance the manuscript's clarity and relevance:

1.       Figure 2 is ambiguous and lacks sufficient descriptions and illustrations. Please provide more details and clarify its content to aid reader understanding.

2.       Please ensure that you have obtained the necessary permissions for reproducing figures from previously published papers and provide appropriate acknowledgments or citations.

3.       The manuscript lacks a comprehensive discussion of the current limitations and challenges in the field, as well as potential solutions. Please expand upon these aspects to provide a well-rounded review.

4.       Please discuss the design criteria for delivery systems in this application, such as size, morphology, surface charge, materials, and their underlying mechanisms. This will provide readers with a better understanding of the factors that influence the effectiveness of these delivery systems.

5.       Please check for recent publications in this research area and update your manuscript accordingly to ensure that your content is up-to-date and comprehensive.

6.       Please discuss the common methods for fabrication for inhalable delivery systems, such as nebulization, and their implications for the design and functionality of these systems.

Author Response

Referee 3:

This manuscript presents a valuable review of the use of lipid-based nanoparticles as inhalable therapeutics for non-small cell lung cancer (NSCLC) treatment. However, several issues need to be addressed to enhance the manuscript's clarity and relevance:

  1. Figure 2 is ambiguous and lacks sufficient descriptions and illustrations. Please provide more details and clarify its content to aid reader understanding.

Authors: We apologise for not being able to make that figure more informative and illustrious. We have added the required description in the image in order to improve clarity.

  1. Please ensure that you have obtained the necessary permissions for reproducing figures from previously published papers and provide appropriate acknowledgments or citations.

Authors: We have cited the references for the images we have used in our paper, and have obtained necessary permissions.

  1. The manuscript lacks a comprehensive discussion of the current limitations and challenges in the field, as well as potential solutions. Please expand upon these aspects to provide a well-rounded review.

Authors: We are thankful to the reviewer for this very important comment, and have introduced a new section entitled “Challenges and Opportunities” in this revised manuscript, which effectively addressed this issue.

  1. Please discuss the design criteria for delivery systems in this application, such as size, morphology, surface charge, materials, and their underlying mechanisms. This will provide readers with a better understanding of the factors that influence the effectiveness of these delivery systems.

Authors: We thank the reviewer for their insightful comment. We have discussed the important parameters that can influence the effectiveness of these inhalable delivery systems.

  1. Please check for recent publications in this research area and update your manuscript accordingly to ensure that your content is up-to-date and comprehensive.

Authors: We have added more recent publications in the paper to make our paper up-to-date. The added references are highlighted in red colour.

  1. Please discuss the common methods for fabrication for inhalable delivery systems, such as nebulization, and their implications for the design and functionality of these systems.

Authors: We have made a new section (Section 2) about types of delivery systems.  

Reviewer 4 Report

The authors have written a useful and comprehensive review of lipid based nanocarriers and their potential promise for treating NSCLC. Several suggestions/edits are listed below:

Line 10 “it’s” should be “its”

Lines 41-42 are outdated; immunotherapy is now standard of care for many lung cancers and additional sentences should be added to reflect new standard of care therapies

Lines 52-53 is somewhat misleading – many of the side effects are directly related to the mechanism of action of the drugs and not the excipient

Line 61 – briefly discuss EPR effect and studies questioning its relevance in humans

Lines 238-240, 333-335, 467-468 – figure legends for Fig 3-5 should be expanded to explain the various panels

Line 263 – should read “explored using a mouse model” or “explored using mice”

Lines 292-294, 513 – replace @ with appropriate symbols on these lines and throughout the manuscript

Line 484 “thy” should be “they”

A short summary should be added regarding potential toxicities and off-target effects of these nanocarriers.

Author Response

Referee 4:

The authors have written a useful and comprehensive review of lipid based nanocarriers and their potential promise for treating NSCLC. Several suggestions/edits are listed below:

Line 10 “it’s” should be “its”

Authors: We have done the necessary correction as suggested.

Lines 41-42 are outdated; immunotherapy is now standard of care for many lung cancers and additional sentences should be added to reflect new standard of care therapies

Authors: We have done the necessary correction as suggested.

Lines 52-53 is somewhat misleading – many of the side effects are directly related to the mechanism of action of the drugs and not the excipient

Authors: We have done the necessary correction as suggested.

Line 61 – briefly discuss EPR effect and studies questioning its relevance in humans

Authors: We have done a brief discussion on EPR as suggested.

Lines 238-240, 333-335, 467-468 – figure legends for Fig 3-5 should be expanded to explain the various panels

Authors: We have explained the figure contents and various panels for Fig 3-5 in the manuscript, the added explanation is highlighted in red colour.

Line 263 – should read “explored using a mouse model” or “explored using mice”

Authors: We have done the necessary correction as suggested.

Lines 292-294, 513 – replace @ with appropriate symbols on these lines and throughout the manuscript

Authors: We have done the necessary correction as suggested.

Line 484 “thy” should be “they”

Authors: We have done the necessary correction as suggested.

A short summary should be added regarding potential toxicities and off-target effects of these nanocarriers.

Authors: We are thankful to the reviewer for this very important comment, and have introduced a new section entitled “Challenges and Opportunities” in this revised manuscript, which effectively addressed this issue.

Reviewer 5 Report

In the manuscript, the authors summarized the application of lipid-based nanoparticles have been developed as both aqueous dispersed as well as dry-powder formulations for inhalable delivery in NSCLC models in vitro and in vivo. However, the innovation of this manuscript needs to be strengthened. So, this manuscript was not enough to be published in Pharmaceutics. Meanwhile, there are some points in this manuscript were unreasonable. The main questions are listed below:

1The inhalable preparations to treat NSCLCs has been reported.(https://doi.org/10.3390/pharmaceutics15010139). This review article focused on the various inhalable formulations for targeted drug delivery, including nano-based delivery systems such as lipids, liposome, polymeric and inorganic nanocarriers, micelles, microparticles and nanoaggregates for lung cancer treatment. Various devices used in pulmonary drug delivery loaded on various nano-carriers are also discussed in detail. This review has described lipid-based nanocarriers for NSCLCs in detail, and there is no significant improvement in this aspect in this manuscript. Therefore, the innovation of this manuscript was not enough.  

2The focus of this manuscript is on lipid-based inhalable micro- and nanocarriers for treating NSCLCs. However, the authors only introduced the advantages of micro- and nanoparticles in introduction, and did not introduce lipid-based micro- and nanocarriers. Please add an introduction about lipid-based micro- and nanocarriers in introduction.

3It is suggested to use a sentence to summarize the content of this manuscript in the last paragraph of the introduction to improve reader’s readability.

4The fourth part of the manuscript introduced the various equipment for the delivery of inhalable therapeutics. However, the authors also introduced the equipment of inhaled medication in the introduction, causing duplication of content. It is suggested to combine the equipment content in the introduction with the fourth part to form the integrity of the article.

5In the second part of this manuscript, the authors only introduced the types of lipid micro- and nanoparticles and the basic knowledge of each type. It is suggested to supplement examples of inhalable preparations used for NSCLCs with each type of lipid nanoparticles in the second part to clarify the characteristics of this preparation.

6In the second part of the manuscript, the author mentioned lipid nanodiscs and nanocubosomes but these different preparation types need to be exemplified in the examples in third part. Please supplement relevant examples.

7In the third part, the authors elaborated on the use of lipid micro- and nanoparticles for chemotherapygene therapy and other therapy. But the main content of the authors was not reflected in the classification title. Please modify and supplement the classification title.

8Section 3.1 of this manuscript listed lipid inhalable preparations for chemotherapy of NSCLCs. The references cited in several examples were older. Please replace them with the latest five years’ literature or explain the significance of using these references.

9The examples listed in section 3.1 of this manuscript were mostly of liposome, Please add other examples of other formulations of lipid-based.

10Section 3.2 of the manuscript listed lipid inhalable preparations for gene therapy, like plasmid and siRNA. Please supplement other types of gene therapy, such as mRNAand shRNA.

11The drugs listed in Table 2 in section 3.2 of the manuscript were chemotherapeutic drugs, and do not fall within the scope of gene therapy. Please integrate the content of Table 2 with 3.1.

12This manuscript mentioned that lipid-based inhalable preparations have great prospects for the treatment of NSCLCs. However, there was no list of lipid inhalable preparations marketed or used in clinical trials for NSCLCs apart from the list of FDA approved liposomes in Part 1. Please provide examples of different marketed lipid-based micro- and nanocarriers based on their trade names, indications, years of marketing, etc.

13In the conclusions part, the authors only mentioned the advantages and prospects of lipid inhalation preparations, but the current shortcomings and possible solutions have not been summarized. Please supplement them.

14Please add a graphical abstract of the manuscript to summarize of the manuscript.

15There were multiple input errors in the manuscript, such as incorrect symbols in "1.2-1.6 m" on line 253. Please check and correct them.

16It is recommended that the English in the manuscript need to be modified.

Author Response

Referee 5:

In the manuscript, the authors summarized the application of lipid-based nanoparticles have been developed as both aqueous dispersed as well as dry-powder formulations for inhalable delivery in NSCLC models in vitro and in vivo. However, the innovation of this manuscript needs to be strengthened. So, this manuscript was not enough to be published in Pharmaceutics. Meanwhile, there are some points in this manuscript were unreasonable. The main questions are listed below:

1、The inhalable preparations to treat NSCLCs has been reported.(https://doi.org/10.3390/pharmaceutics15010139). This review article focused on the various inhalable formulations for targeted drug delivery, including nano-based delivery systems such as lipids, liposome, polymeric and inorganic nanocarriers, micelles, microparticles and nanoaggregates for lung cancer treatment. Various devices used in pulmonary drug delivery loaded on various nano-carriers are also discussed in detail. This review has described lipid-based nanocarriers for NSCLCs in detail, and there is no significant improvement in this aspect in this manuscript. Therefore, the innovation of this manuscript was not enough. 

Authors: We have read the published paper, and have tried our best to make our review more unique through exhaustive revisions in this re-submission.

2、The focus of this manuscript is on lipid-based inhalable micro- and nanocarriers for treating NSCLCs. However, the authors only introduced the advantages of micro- and nanoparticles in introduction, and did not introduce lipid-based micro- and nanocarriers. Please add an introduction about lipid-based micro- and nanocarriers in introduction.

Authors: We have included a section (Section 3) about major lipid-based micro/nanocarriers, including natural carriers such as exosomes.

3、It is suggested to use a sentence to summarize the content of this manuscript in the last paragraph of the introduction to improve reader’s readability.

Authors: We have summarized the content of the manuscript in the last paragraph of the introduction section.

4、The fourth part of the manuscript introduced the various equipment for the delivery of inhalable therapeutics. However, the authors also introduced the equipment of inhaled medication in the introduction, causing duplication of content. It is suggested to combine the equipment content in the introduction with the fourth part to form the integrity of the article.

Authors: We have removed the duplication by incorporating such information only in Section 2.

5、In the second part of this manuscript, the authors only introduced the types of lipid micro- and nanoparticles and the basic knowledge of each type. It is suggested to supplement examples of inhalable preparations used for NSCLCs with each type of lipid nanoparticles in the second part to clarify the characteristics of this preparation.

Authors: Although several types of lipid-based micro/nanocarriers exist, as elaborated in section 3, not all of them have been used in inhalable drug delivery against NSCLC (though many have been used for inhalable delivery against other lung diseases, such as cystic fibrosis). Therefore, such examples could not be incorporated. As such, inhalable delivery against NSCLC is quite a nascent field, and other types of lipid-based inhalable nanocarriers should be explored against this disease. We have mentioned this point in the newly introduced ‘Challenges and opportunities’ section in this revised manuscript.  

6、In the second part of the manuscript, the author mentioned lipid nanodiscs and nanocubosome, but these different preparation types need to be exemplified in the examples in third part. Please supplement relevant examples.

Authors: Among these two particular lipid-based nanovesicles mentioned, we have found only one example (using cubosomes) of inhalable drug delivery against NSCLC. This report has been included in this revised manuscript.

7、In the third part, the authors elaborated on the use of lipid micro- and nanoparticles for chemotherapy、gene therapy and other therapy. But the main content of the authors was not reflected in the classification title. Please modify and supplement the classification title.

Authors: We apologise for being not able to understand this comment. We shall be glad to make necessary corrections if we receive specific directions from the learned reviewer about this comment.  

8、Section 3.1 of this manuscript listed lipid inhalable preparations for chemotherapy of NSCLCs. The references cited in several examples were older. Please replace them with the latest five years’ literature or explain the significance of using these references.

Authors: We thank the reviewer for their valuable comment. We agree that several of the references cited in Section 4 are older than five years. We have done so to present an overview of the research done on this topic in the last two decades and discuss the pioneering and noticeable experimental work that has been done. In order to keep the paper up-to-date, we have added latest references of this field in Table 2.  

9、The examples listed in section 3.1 of this manuscript were mostly of liposome, Please add other examples of other formulations of lipid-based.

Authors: The literature shows most examples about inhalable drug delivery for treating NSCLC involved liposomes. There are very few examples of other types of lipid based carriers for inhalable treatment of NSCLC, though other lung diseases, such as cystic fibrosis, has been treated. We have added few examples of other lipid-based nanocarriers (e.g. using nanocubosomes) in this revised manuscript. In the newly introduced ‘Challenges and opportunities’ section, we have discussed the need for exploring other types of lipid-based nanoformulations for inhalable treatment of NSCLC.  

10、Section 3.2 of the manuscript listed lipid inhalable preparations for gene therapy, like plasmid and siRNA. Please supplement other types of gene therapy, such as mRNA,and shRNA.

Authors: Reports on lipid-based inhalable delivery of several forms of DNA/RNA against NSCLC are not available in literature, though such reports exist against the treatment of other lung disorders, such as cystic fibrosis (CF). We have included an example of mRNA delivery against CF in this revised manuscript. Also, a report containing the design and development of a shRNA co-delivery formulation for DPI-based delivery against lung cancer is included.  

11、The drugs listed in Table 2 in section 3.2 of the manuscript were chemotherapeutic drugs, and do not fall within the scope of gene therapy. Please integrate the content of Table 2 with 3.1.

Authors: We have added some examples of gene therapy in Table 2.

12、This manuscript mentioned that lipid-based inhalable preparations have great prospects for the treatment of NSCLCs. However, there was no list of lipid inhalable preparations marketed or used in clinical trials for NSCLCs apart from the list of FDA approved liposomes in Part 1. Please provide examples of different marketed lipid-based micro- and nanocarriers based on their trade names, indications, years of marketing, etc.

Authors: We thank the reviewer for this comment, we have added several lipid- based formulations that are undergoing clinical trials for the treatment of NSCLCs in Section 3.

13、In the conclusions part, the authors only mentioned the advantages and prospects of lipid inhalation preparations, but the current shortcomings and possible solutions have not been summarized. Please supplement them.

Authors: We are thankful to the reviewer for this very important comment, and have introduced a new section entitled “Challenges and Opportunities” in this revised manuscript, which effectively addressed this issue.

14、Please add a graphical abstract of the manuscript to summarize of the manuscript.

Authors: We have introduced a graphical abstract, as suggested.

15、There were multiple input errors in the manuscript, such as incorrect symbols in "1.2-1.6 m" on line 253. Please check and correct them.

Authors: We have done the necessary corrections as suggested.

16、It is recommended that the English in the manuscript need to be modified.

Authors: We have carried out a more rigorous proof-reading to improve the English in this manuscript.

Round 2

Reviewer 1 Report

The authors have corrected the manuscript as per my comments and suggestion. Now, the manuscript looks better, thus recommend it for publication.

Author Response

Authors: We sincerely thank the Reviewer for the encouraging comment.

Reviewer 2 Report

I have no further questions.

Author Response

(The authors gave the same response as above.)

Reviewer 5 Report

Recommendation: Minor repair

The authors have made great efforts to address our comments in the revised manuscript. However, only question still existed and was not fully answered. Detailed suggestion is listed below:

For question 7, we apologize for not expressing it well enough for the authors to understand our meaning.In the fourth part, the author describes the application of lipid particles and nanoparticles in chemotherapy, gene therapy and other treatments. However, the subtitles of fourth part were "Lipid nanoparticles for inhalable drug delivery", "Lipid nanoparticles for gene delivery to lungs", and "Other therapies". Among them, the " Lipid nanoparticles for inhalable drug delivery " includes " Lipid nanoparticles for gene delivery to lungs " and " Other therapies ". We suggest that the authors change section 4.1 to reflect "chemotherapy" in the subtitles and improve the corresponding content.

Author Response

Authors: We sincerely thank the Reviewer for the encouraging comment. We have corrected the heading of Section 4.1, as suggested. Further, we have improved the English Language (spelling and grammer), as indicated by the Reviewer.